# Fast-decaying plant litter enhances soil carbon in temperate forests but not through microbial physiological traits

Matthew E. Craig [1,2✉], Kevin M. Geyer [3,4], Katilyn V. Beidler [1], Edward R. Brzostek [5], Serita D. Frey [3], A. Stuart Grandy[3], Chao Liang[6] & Richard P. Phillips [1]

Conceptual and empirical advances in soil biogeochemistry have challenged long-held assumptions about the role of soil micro-organisms in soil organic carbon (SOC) dynamics; yet, rigorous tests of emerging concepts remain sparse. Recent hypotheses suggest that microbial necromass production links plant inputs to SOC accumulation, with high-quality (i.e., rapidly decomposing) plant litter promoting microbial carbon use efficiency, growth, and turnover leading to more mineral stabilization of necromass. We test this hypothesis experimentally and with observations across six eastern US forests, using stable isotopes to measure microbial traits and SOC dynamics. Here we show, in both studies, that microbial growth, efficiency, and turnover are negatively (not positively) related to mineral-associated SOC. In the experiment, stimulation of microbial growth by high-quality litter enhances SOC decomposition, offsetting the positive effect of litter quality on SOC stabilization. We suggest that microbial necromass production is not the primary driver of SOC persistence in temperate forests. Factors such as microbial necromass origin, alternative SOC formation pathways, priming effects, and soil abiotic properties can strongly decouple microbial growth, efficiency, and turnover from mineral-associated SOC.

[1] Department of Biology, Indiana University, Bloomington, IN, USA. [2] Environmental Sciences Division and Climate Change Science Institute, Oak Ridge National Laboratory, Oak Ridge, TN, USA. [3] Department of Natural Resources and the Environment, University of New Hampshire, Durham, NH, USA. [4] Department of Biology, Young Harris College, Young Harris, GA, USA. [5] Department of Biology, West Virginia University, Morgantown, WV, USA. [6] Key Laboratory of Forest Ecology and Management, Institute of Applied Ecology, Chinese Academy of Sciences, Shenyang, China. ✉email: craigme@ornl.gov

The conversion of plant inputs into stable soil organic carbon (SOC) represents a critical yet poorly understood process, with consequences for soil C storage, nutrient availability, net primary production, and ecosystem sensitivity to global change[1–4]. The largest and slowest-cycling SOC pool is largely composed of microbial products stabilized by associations with soil minerals[5–7], suggesting that microbial production mediates the transfer of plant inputs into the mineral-associated SOC pool. Microbial necromass (dead microbial cells and their degradates) accounts for a large portion of SOC in many systems[8,9] and necromass production is controlled by three physiological traits: microbial growth rate (MGR), microbial carbon use efficiency (CUE)—the proportion of assimilated microbial C allocated to the production of new biomass—and microbial biomass turnover rate (MTR). Though these traits may be central to both SOC formation and decomposition[10–12], the controls on microbial physiological traits and their consequences for SOC across environmental gradients are poorly elucidated, hindering predictions about how SOC pools will respond to environmental changes.

Contemporary SOC theory predicts that microbial physiological traits mediate SOC stabilization, with greater microbial growth, efficiency, or turnover leading to greater mineral-associated SOC[10] (hereafter the 'necromass stabilization hypothesis'; Fig. 1). Multiple isotope tracing studies support that high-quality, fast-decomposing plant inputs are more rapidly or efficiently transferred into mineral-associated SOC[1,13–18], but see[19]. This effect is commonly attributed to the necromass stabilization hypothesis, assuming that higher input quality promotes faster or more efficient microbial growth[10,20,21]. The necromass stabilization hypothesis is represented implicitly in conventional first-order decay models[22], and is increasingly represented in more mechanistic microbially explicit SOC models, which often treat microbial necromass as a primary source of mineral-associated SOC[11,12,22–28]. Yet, despite its importance in conceptual and mathematical models, this hypothesis has only limited empirical support (e.g., from artificial laboratory microcosms[29] or single-site field observations[30]). Moreover, other mechanisms could lessen the importance of the necromass stabilization pathway—offsetting necromass buildup via enhanced SOC turnover (e.g., priming effects)[31–33] or circumventing microbial physiological traits via non-necromass SOC formation pathways (e.g., direct sorption of plant compounds[34,35] or microbial extracellular products[36,37]; Fig. 1a). Such mechanisms are rarely considered alongside the necromass stabilization pathway in empirical studies.

Here, we tested a hypothesis that is central to current SOC models—that microbial physiological traits link litter quality to SOC stabilization (H1; Fig. 1a). We used both experimental microcosms and a multi-site field study to ask whether microbial physiological traits (1) predict differences in mineral-associated SOC and (2) represent the key link between plant litter decomposition and SOC stabilization. We evaluated competing hypotheses (Fig. 1a) that direct interactions between plant compounds and mineral surfaces could explain litter quality effects on mineral-associated SOC (H2), microbial extracellular production could decouple microbial necromass production from mineral-associated SOC formation (H3), and microbial growth and efficiency stimulation could promote decomposition and reduce net SOC gains from litter-enhanced SOC formation (H4). In the microcosm experiment, we incubated 16 different temperate tree leaf litters with isotopically distinct soil and measured microbial physiological traits and the flow of litter-derived C into the mineral-associated SOC pool. Given that the characteristics of the soil matrix (e.g., soil texture, pH, metal-oxides) may exert more control over mineral-associated SOC retention than the source or characteristics of the organic matter itself[10,38,39] (H5), we also

tested hypotheses about the relationship between plant litter, microbes, and edaphic factors (Fig. 1a) across six eastern US temperate forests (Fig. 1b). We quantified SOC pools (mineral-associated and necromass-derived) along with microbial physiological traits and other potential biotic and abiotic drivers of mineral-associated SOC. We find that alternative SOC formation

**a**

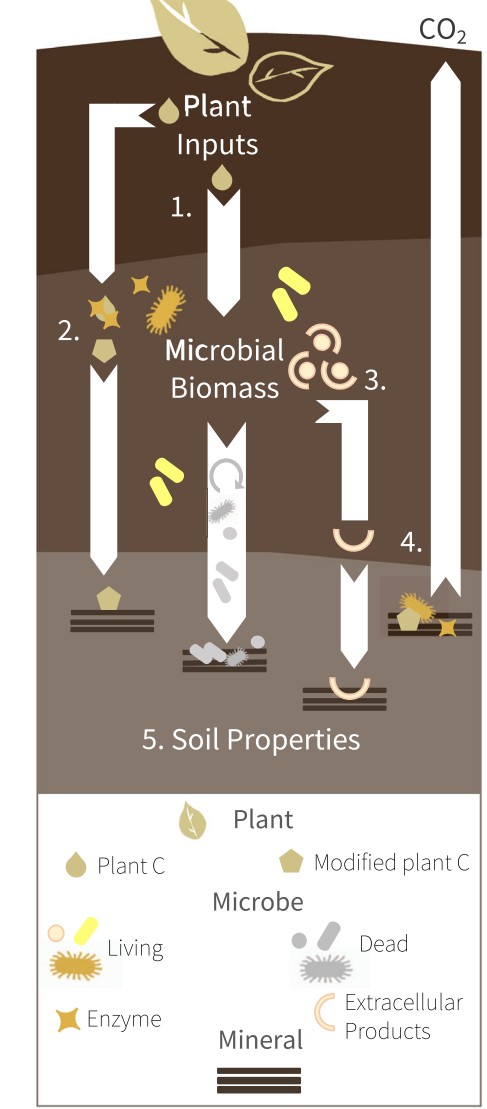

**b**

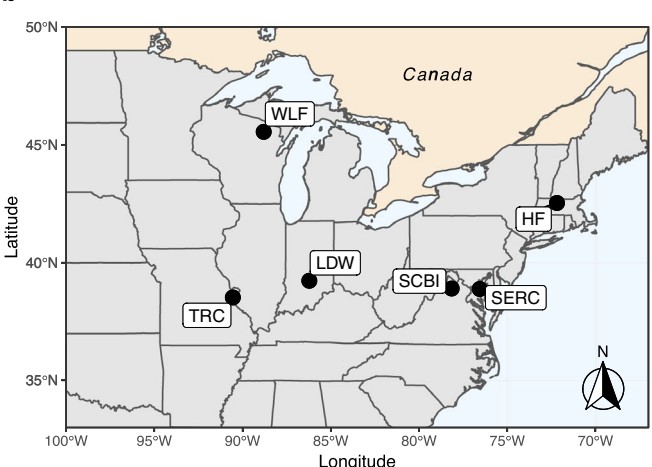

**Fig. 1 Conceptual model and map of study sites. a** Conceptual model showing microbe-mediated mechanisms of mineral-associated soil organic carbon (SOC) formation and decay. Pathway 1 represents the necromass stabilization hypothesis and we note that different types of necromass may be differentially susceptible to mineral stabilization. Other numbers represent (2) mineral stabilization of plant inputs without assimilation by microbes, (3) mineral stabilization of microbial extracellular compounds, (4) stimulation of microbe-mediated mineral-associated SOC decay, (5) and the role of soil properties in governing mineral-associated SOC accumulation. **b** Map of study sites including Wabikon Lake Forest (WLF), Harvard Forest (HF), Lilly-Dickey Woods (LDW), Smithsonian Conservation Biology Institute (SCBI), Smithsonian Environmental Research Center (SERC), and Tyson Research Center (TRC).

pathways and priming effects act to decouple microbial physiological traits from SOC formation in temperate forests.

## Results

**Microcosm experiment**. We monitored litter decomposition by measuring the rate and isotopic signature of respiration during 185-day incubations of litter-soil mixtures. Litter treatments varied markedly in chemical traits and decomposition parameters (Table S1). A litter quality index, obtained via a principal components analysis (Fig. S1A), explained 47% of this variation and correlated highly with acid unhydrolyzable residue (AUR) content—an indicator of litter carbon quality[40]. We used a substrate-nonspecific technique ($^{18}O$ incorporation into DNA) to quantify microbial physiological traits during early (15 days) and intermediate (100 days) stages of decomposition (Table S2). These time intervals correspond to the midpoint of parallel early and intermediate carbon budget microcosms (described below). Microbial growth, CUE, and turnover were highly correlated with one another allowing us to collapse these traits into a single microbial physiological trait index explaining 90% of the variation (Fig. S1B).

High-quality litters increased microbial growth, CUE, and turnover (Fig. 2a). This effect was especially pronounced during intermediate stages of decomposition suggesting a depletion of the smaller pool of labile substrates in low-quality litters. Microbial physiological traits were better predicted by indicators of litter C quality than by litter nitrogen (N) content (Fig. S2), which is notable given that N availability and substrate stoichiometry are considered primary drivers of microbial physiological traits[41,42]. Nonetheless, the positive effect of litter quality on microbial physiological traits supports the common hypothesis that high-quality or fast-decomposing litters should promote rapid and efficient microbial growth[10]. Whereas this hypothesis has been previously evaluated indirectly by tracing microbial use of specific C compounds[10,22], these substrate non-specific measurements provide direct evidence from microbial communities growing on natural leaf litter substrates.

Contrary to the predictions of the necromass stabilization hypothesis, microbial physiological traits were not positively related to the amount of litter C recovered in the mineral-associated SOC pool during early (30 days) or intermediate (185 days) stage decomposition (Fig. 2b). This is surprising given that high-quality litters led to faster and more efficient mineral-associated SOC formation (Fig. S3), a common finding that is often attributed to the necromass stabilization hypothesis. Instead, path analysis revealed a negative relationship between microbial physiological traits and both the rate and efficiency of mineral-associated SOC formation (Fig. 2c; Fig. S4). This pattern was most pronounced in intermediate stages when the effects of litter quality on microbial physiological traits were strongest. The

overall relationship between litter quality and mineral-associated SOC formation was explained by a direct linkage rather than by an indirect effect—that is, it was not mediated through microbial physiological traits. In fact, the indirect microbial physiological trait effect counteracted and weakened the overall positive effect of litter quality on mineral-associated SOC formation in both early and, especially, intermediate stages of decomposition.

These findings point toward a few explanations with important implications for conceptual and mathematical SOC models. The inability of microbial physiological traits to explain litter quality effects on mineral-associated SOC formation suggests that the production of microbial necromass is not a sufficient explanation for SOC stabilization in temperate systems. This result highlights the critical importance of non-necromass pathways of mineral-associated SOC formation. The direct linkage between litter quality and mineral-associated SOC formation in our path analysis could indicate direct sorption of plant-derived decomposition products[14,31,43] (Fig. 1a). Indeed, high-quality litters in this experiment had more soluble material and decomposed faster, likely enhancing dissolved SOC[44] and producing oxidized intermediates with a high sorption affinity[35]. These effects of litter quality on dissolved SOC are known to enhance mineral-associated SOC formation[45]. In addition, microbial extracellular products could be important (Fig. 1a), but are overlooked by an emphasis on microbial physiological traits[36,46]. Extracellular polymeric substances, stress compounds, and similar products account for a small proportion of microbial production[47], but are likely to disproportionately affect mineral-associated SOC given their susceptibility to stabilization on mineral surfaces[37].

In direct contrast with predictions derived from the necromass hypothesis, litter quality stimulation of microbial physiological traits may also mediate priming effects (i.e., stimulation of SOC decay by fresh inputs) which act to reduce, rather than promote, mineral-associated SOC formation. Higher litter quality promoted faster and more efficient microbial growth, leading to increased microbial biomass (Fig. S5) which is likely to promote SOC decomposition via increased microbial activity[48]. Though mineral-associated SOC is often assumed to resist microbial decomposition, a large fraction of this pool is likely susceptible to microbe-mediated desorption[2]. Microbes can access mineral-associated organic matter as a source of N and microbial activity can promote desorption by altering redox states or diffusion gradients, or by producing extracellular enzymes[49]. Alternatively, newly produced necromass could be recycled as a source of N prior to mineral stabilization[50]. Thus, the negative relationship between microbial physiological traits and mineral-associated SOC formation in the path analysis indicates that decomposition of newly formed mineral-associated SOC or mineral-associated SOC precursors outweighed necromass-driven SOC formation. Though rarely assessed[32,51], the balance of these two processes is likely to vary based on the strength of mineral-organic associations and magnitude of litter-induced priming effects. This balance should determine whether the necromass pathway is a dominant driver of mineral-associated SOC formation in a given system.

We observed a negative effect of microbial physiological traits on soil-derived (i.e., pre-existing) mineral-associated SOC (Fig. 3a) and particulate SOC (Fig. S6), supporting the hypothesis that litter quality stimulation of microbial physiological traits led to priming effects. By the time intermediate-stage decomposition was reached, this led to a significant negative indirect effect of litter quality on soil-derived SOC. A litter-induced priming effect was also evident in cumulative soil-derived respiration data, but we note that this was better predicted by the second axis of the litter quality index—which reflected litter N content (Fig. 3b)—potentially indicating that SOC formation and decomposition

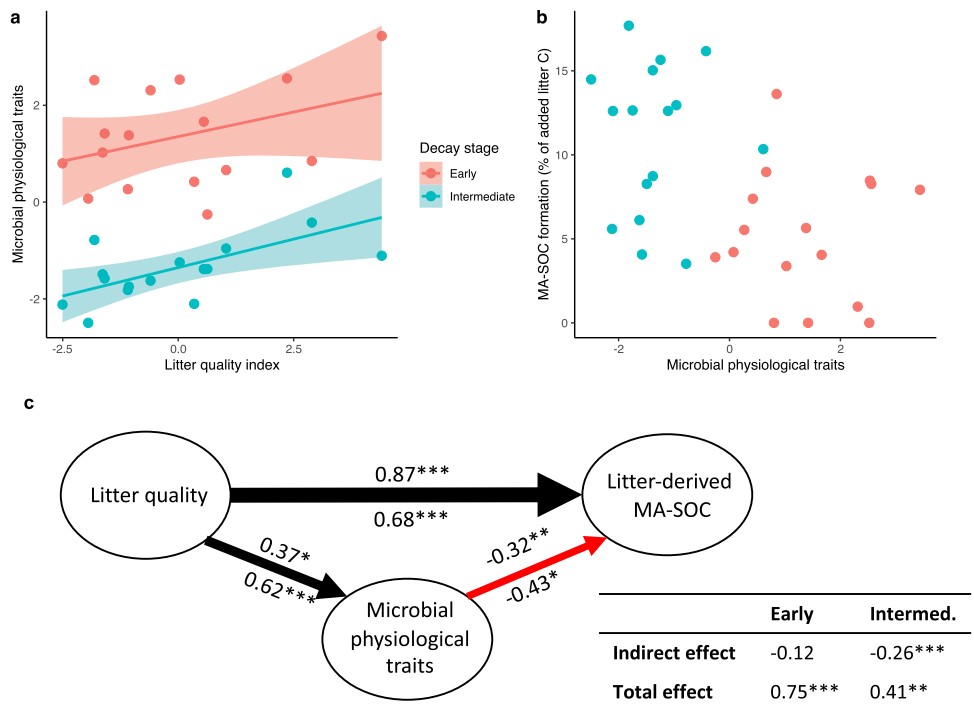

**Fig. 2 Microbial physiological traits and mineral-associated SOC (MA-SOC) formation in the microcosm experiment. a** Linear relationship (±SE; $n = 16$) between the litter quality index (PC1 in Fig. S1A) and the microbial physiological trait index (PC1 in Fig. S1B) after 15 days (Early stage: $R^2 = 0.14$; $P = 0.16$) and 100 days (Intermediate stage: $R^2 = 0.38$; $P = 0.01$) of decomposition, (**b**) lack of a relationship ($P > 0.96$) between the microbial physiological trait index and litter-derived MA-SOC after 30 (Early) and 185 days (Intermediate), and (**c**) path analysis showing the direct and indirect effects of the litter quality index (Litter quality) on the percentage of added litter C recovered in mineral-associated SOC (Litter-derived MA-SOC). Indirect effects of litter quality are mediated through the microbial physiological trait index. Numbers above and below paths represent standardized coefficients during early- and intermediate-stage decomposition, respectively, with significance levels indicated (*$p < 0.1$, **$p < 0.05$, and ***$p < 0.01$). Thickness and color of lines correspond to coefficient magnitude and direction, respectively. Total and indirect effects of litter quality on soil C formation are also summarized with standardized coefficients.

were differentially affected by litter C quality vs. N availability. In summary, our microcosm experiment suggests that the positive effect of litter quality on mineral-associated SOC formation is attributable to the sorption of litter-derived or microbial extracellular products rather than microbial necromass. To the contrary, elevated microbial growth, efficiency, and turnover may drive the decomposition of new and old SOC.

**Field study**. To evaluate whether the mechanisms implied by our experimental results are relevant drivers of mineral-associated SOC in natural forests, we quantified microbial physiological traits along with mineral-associated and necromass-derived SOM pools across wide environmental gradients nested within six US temperate forests. These gradients were identified based on the mycorrhizal associations of dominant trees[52–54] (see methods) and reflected variation in litter quality and other factors hypothesized to control microbial physiological traits (e.g., soil C:N, pH, and N availability; Table S3). As in the microcosm experiment, we derived a litter quality index and a microbial physiological trait index explaining 45% of the variation in litter chemical traits and 79% of the variation in microbial physiological traits, respectively (Fig. S1C, D).

In contrast with the microcosm experiment, we detected no relationship between litter quality and microbial physiological traits in the field (Fig. S7), which is perhaps unsurprising given field variation in other factors expected to control litter decomposition and microbial functioning (e.g., soil nutrient availability). Microbial physiological traits were instead positively

predicted by soil C:N (Fig. S7), meaning microbial growth, efficiency, and turnover tended to be greater in soils with more carbon per unit nitrogen. This indicates that our plots, as well as the soil used in our experiment (C:N = 12.0), fell below the stoichiometric boundary (i.e., threshold element ratio) between C vs. N limitation[55], which is thought to range from a C:N of 20–25 for terrestrial microbial communities[56]. Indeed, bivariate plots suggest a hump-shaped relationship where microbial physiological traits no longer increase with soil C:N at a soil C:N of ~20 (Fig. S8), suggesting that N limitation might drive microbial physiological traits in soils with a higher C:N ratio.

Patterns in mineral-associated SOC, however, were consistent with the microcosm experiment. Litter quality was positively associated with mineral-associated SOC, but this relationship was not explained by microbial physiological traits (Fig. 4a; Fig. S9), which were unrelated to total mineral-associated SOC (Fig. S9). Consistent with the microcosm experiment, we observed a negative relationship between microbial physiological traits and the proportion of soil C stored in mineral-associated SOC (Fig. 4a). This pattern may have been driven by alternative stabilization pathways or priming of mineral-associated SOC, as documented in the microcosm experiment, but our observations reveal additional mechanisms by which necromass production could be decoupled from mineral-associated SOC along natural gradients.

Differences in necromass stabilization could weaken the relationship between necromass production and accumulation across field plots. We found no significant relationship between microbial physiological traits and the size of the soil microbial

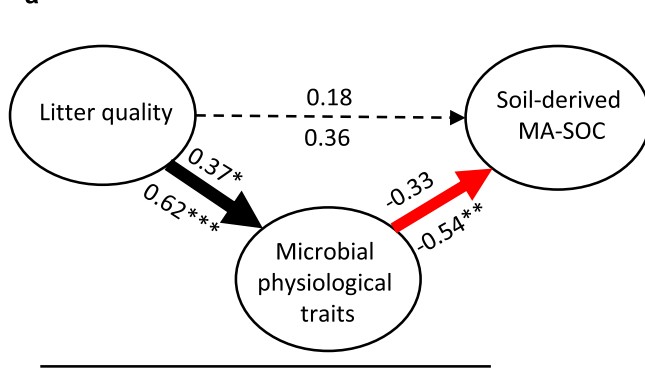

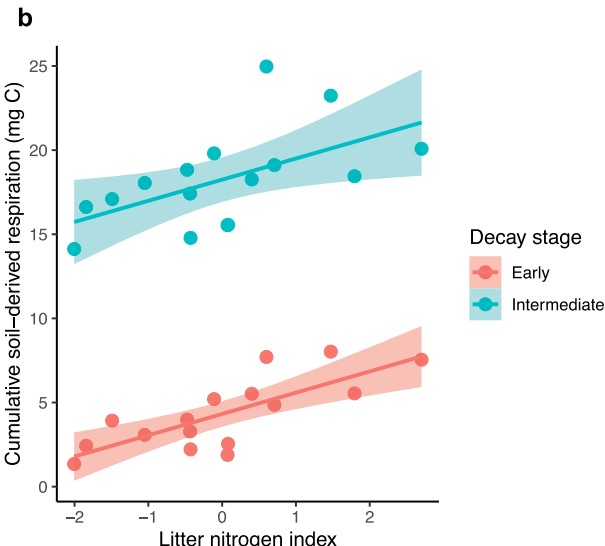

**Fig. 3 Soil-derived (i.e., pre-existing) carbon losses in the microcosm experiment. a** Path analysis showing the direct and indirect effects of the litter quality index (Litter quality) on soil-derived mineral-associated SOC (MA-SOC). Indirect effects of litter quality are mediated through the microbial physiological trait index. Numbers above and below paths represent standardized coefficients during early- and intermediate-stage decomposition, respectively, with significance levels indicated (*$p < 0.1$, **$p < 0.05$, and ***$p < 0.01$). Thickness and color of lines correspond to coefficient magnitude and direction, respectively. Total and indirect effects of litter quality on soil C formation are also summarized with standardized coefficients. **b** Linear relationship (±SE; $n = 16$) between the litter nitrogen index—i.e., the second axis of the litter quality PCA which correlated negatively with litter C:N and AUR:N (Fig. S1A)—and cumulative soil-derived respiration ($CO_2$-C) after 30 days (Early: $R^2 = 0.59$; $P < 0.01$) and 185 days of decomposition (Intermediate: $R^2 = 0.32$; $P = 0.02$).

necromass pool obtained via soil amino sugar analyses of the field soils (Fig. 4b), suggesting that necromass production was not a sufficient predictor of necromass retention. We observed only a weak positive relationship of total necromass with $Fe_{ox}$ and, curiously, a negative relationship with soil clay content, whereas the necromass stabilization hypothesis assumes that microbial necromass is highly susceptible to interactions with soil minerals. Recent work, however, suggests necromass retention may be more influenced by substrate chemistry than by mineral mechanisms in fungal-dominated forest topsoils[57,58], potentially leading to the buildup of necromass in organic soils or particulate organic matter fractions[59]. After resolving necromass into fungal and bacterial pools, it is clear that fungal necromass drove the negative association with clay content—likely reflecting fungal dominance in coarse-textured soils[60]—while bacterial necromass was strongly associated with $Fe_{ox}$ (Fig. S10). This suggests, in agreement with previous work[58,61–64], that bacterial cellular products were more prone to interactions with soil minerals and therefore a more important source of mineral-associated SOC. We suggest that the origin of microbial necromass influences its stabilization potential and therefore the extent to which necromass production should drive mineral-associated SOC formation (Fig. 1a).

Variation in abiotic soil properties could also directly or indirectly override the effect of microbial physiological traits on mineral-associated SOC. Across our plots, clay content was strongly associated with the proportion of soil C stored in mineral-associated SOC and oxalate-extractable Fe ($Fe_{ox}$)—an indicator of poorly crystalline Fe oxides—was the strongest predictor of total mineral-associated SOC (Fig. 4a; Fig. S7). In addition to these direct relationships, abiotic soil properties could indirectly decouple microbial physiological traits from mineral-associated SOC through their association with other SOM and biotic properties. For example, while clay content was positively associated with mineral-associated SOC, clayey soils can have a lower root density[65]. This pattern could partially explain the negative relationships between mineral-associated SOC and root biomass or microbial physiological traits (Fig. 4a)—because roots can promote microbial activity—or the negative relationship between clay content and microbial necromass (Fig. 4b). Indeed, we found that greater fine root biomass was associated with higher microbial biomass (mixed model: $P = 0.09$), necromass (Fig. 4b), and a tendency toward higher microbial physiological traits (Fig. S7).

Roots and root-associated microbes are likely to drive patterns in SOC dynamics. Rhizospheres are important zones of both SOC formation[43] and decay[66]. Our finding that fine root biomass was associated with greater microbial activity and lower mineral-associated SOC could reflect root-driven SOC priming[67]. Yet our study targeted aboveground inputs and upper topsoil; a different role of roots may be apparent at greater depths where belowground inputs dominate. Indeed, root-derived carbon was recently found to account for a substantial and rapid accumulation of mineral-associated SOC in ingrowth cores deployed in these same plots[68], indicating a role of living root or mycorrhizal inputs[69]. Measurements of in situ interactions among roots, microbes, and SOC would be valuable in further improving our understanding of pathways of SOC formation and decomposition.

While we found that microbial growth, efficiency, and turnover are not adequate predictors of mineral-associated SOC formation in surface soils of temperate forests, necromass stabilization on mineral surfaces could still be an important mechanism of SOC storage. This mechanism may explain more variation in other systems (e.g., croplands or grasslands) or in deeper soils, where necromass accounts for a larger proportion of total SOC[9]. For example, agricultural soils tend to be more bacterially dominated[9], potentially leading to stronger interactions between microbial necromass and soil minerals (as described above). This may lead to a stronger association between microbial physiological traits and mineral-associated SOC in cropped systems[30]. Future sampling campaigns are needed to understand how SOC formation pathways vary across different systems.

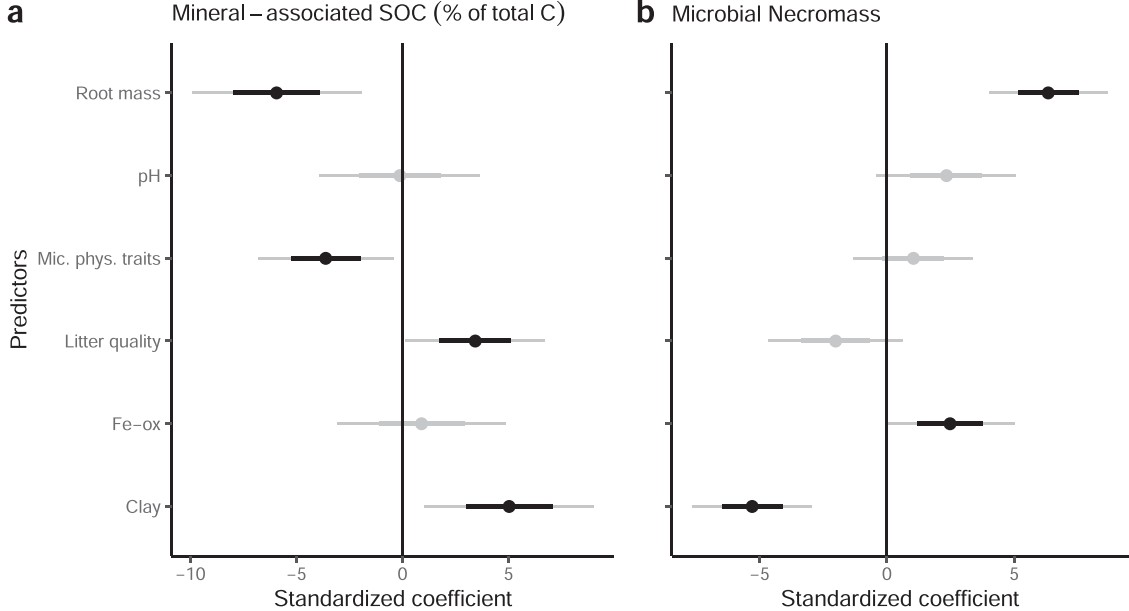

**Fig. 4 Results from the field study.** Linear mixed model coefficients relating the percentage of soil C stored in mineral-associated SOC (**a**) and total microbial necromass (**b**) to the litter quality index (Litter quality; PC1 in Fig. S1C) the microbial physiological trait index (Mic. phys. traits; PC1 in Fig S1D), fine root biomass, soil pH, oxalate-extractable iron (Fe-ox), and soil clay content. Plot shows standard error ($n = 54$; inner bold lines) and 95% confidence intervals (outer lines). Coefficients were centered and standardized to show the relative importance of each predictor despite the different scales on which the variables were measured. Black points indicate $p < 0.1$.

## Discussion

Contemporary SOC concepts posit that microbial physiological traits are key mediators in the mineral stabilization of plant inputs, leading to the hypothesis that high-quality plant inputs promote mineral-associated SOC by enhancing microbial growth and turnover. We found that plant litter quality was positively associated with mineral-associated SOC in both a microcosm experiment and across six temperate forests. This pattern, however, was not explained by microbial physiological traits in either the experiment or the field study. Though litter quality was positively associated with microbial physiological traits in the microcosm experiment, field variation in microbial growth, efficiency, and turnover was better explained by soil C:N ratios. More importantly, microbial physiological traits were negatively related to both mineral-associated SOC formation and the proportion of soil C stored in mineral-associated SOC in the experiment and field study, respectively. Thus, while acknowledging that microbial necromass represents an important pool of SOC, we conclude that the necromass stabilization hypothesis is insufficient as a general explanation linking plant input quality to mineral-associated SOC formation.

Our results highlight four factors which act to decouple microbial necromass production from mineral-associated SOC (Fig. 1a): differences in the mineral stabilization potential of necromass produced by different microbial communities (1); alternative pathways linking plant inputs to mineral-associated SOC (i.e., direct stabilization of plant inputs or the stabilization of microbial extracellular products) (2 and 3); priming of newly formed or pre-existing mineral-associated SOC driven by higher litter quality and greater microbial activity (4); and the overriding effects of soil abiotic properties (5). The balance of these factors is likely context-dependent, but is critical to our predictive understanding of mineral-associated SOC dynamics and therefore atmosphere-soil C feedbacks. As models and experiments probe these alternative hypotheses, a broadened view of how microbes mediate mineral-associated SOC formation should ultimately lead to greater confidence in soil C projections.

## Methods

**Microcosm preparation and incubation.** Leaf litters were collected from Lilly-Dickey Woods, a mature eastern US temperate broadleaf forest located in South-Central Indiana (39°14′N, 86°13′W) using litter baskets and surveys for freshly senesced litter as described in Craig et al.[52]. Of the 19 species collected in Craig et al. (2018), we selected litter from 16 tree species with the goal of maximizing variation in litter chemical traits (Table S1). Litters were air-dried and then homogenized and fragmented such that all litter fragments passed a 4000 μm, but not a 250 μm mesh. Whereas leaf litters had a distinctly C3 $\delta^{13}C$ signature of $-30.1 \pm 1.5$ (mean, standard deviation), we used a $^{13}C$-rich ($\delta^{13}C = -12.6 \pm 0.4$) soil obtained from the A horizon of a 35-yr continuous corn field at the Purdue University Agronomy Center for Research and Education near West Lafayette, Indiana (40°4′N, 86°56′W). The soil is classified as Chalmers silty clay loam (a fine-silty, mixed, superactive, mesic Typic Endoaquoll). Prior to use in the incubation, soils were sieved (2 mm) and remaining recognizable plant residues were thoroughly picked out. Soils were mixed with acid-washed sand—30% by mass—to facilitate litter mixing (see below) and to increase the soil volume for post-incubation processing. The resulting soil had a pH of 6.7 and a C:N ratio of 12.0.

We constructed the experimental microcosms by mixing the 16 litter species with the $^{13}C$-enriched soil. Each litter treatment was replicated four times in four batches (i.e., 16 microcosms per species, 272 total microcosms including 16 soil-only controls). Two batches (C budget microcosms) were used to monitor $CO_2$ efflux and to track litter-derived C into SOM pools, and two batches were used to quantify microbial biomass dynamics.

Incubations were carried out in 50 mL centrifuge tubes modified with an O-ring to prevent leakage and a rubber septum to facilitate headspace sampling. To each microcosm, we added 5 g dry soil, adjusted moisture to 65% water-holding capacity, and pre-incubated for 24 h in the dark at 24 °C. Using a dissecting needle, 300 mg of leaf litter were carefully mixed into treatment microcosms and controls were similarly agitated. This corresponds to an average C addition rate of $27.1 \pm 1.1$ g C kg$^{-1}$ dry soil among the 16 species. During incubation, microcosms were loosely capped to retain moisture while allowing gas exchange, and were maintained at 65% water-holding capacity by adding deionized water every week.

**Carbon budget in microcosms.** Respiration was quantified with an infrared gas analyzer (LiCOR 6262, Lincoln, NE, USA) coupled to a sample injection system. Our first measurement was taken about 12 h after litter addition (day 1) and subsequent measurements were taken on days 2, 4, 11, 19, and 30 for both batches and on days 46, 64, 79, 92, 109, 128, 149, and 185 for the second batch. Prior to each measurement, microcosms were capped, flushed with $CO_2$-free air, and incubated for 1–8 h depending on the expected efflux rate. Headspace was sampled with a gas-tight syringe and the $CO_2$-C concentration was converted to a respiration rate (μg $CO_2$-C day$^{-1}$). Total cumulative $CO_2$-C loss was derived from point measurements by numerical integration (i.e., the trapezoid method). To evaluate soil-derived $CO_2$-C efflux, we measured $\delta^{13}C$ in two gas samples per litter

type or control on a ThermoFinnigan DELTA Plus XP isotope ratio mass spectrometer (IRMS) with a GasBench interface (Thermo Fisher Scientific, San Jose, CA). Isotopes were measured on days 1, 4, 11, 30, 64, 109, and 185. On each of these days, a two-source mixing model[70] was applied to determine the fraction of total $CO_2$-C derived from soil organic matter vs. litter:

$$\frac{F^l(t)}{F(t)} = \frac{\delta F(t) - \delta F^c(t)}{\delta C_l - \delta F^c(t)} \tag{1}$$

where $\frac{F^l(t)}{F(t)}$ is the fraction total $CO_2$-C efflux [$F(t)$] derived from litter [$F^l(t)$] at time ($t$), $\delta F(t)$ is the $\delta^{13}C$ of the $CO_2$ respired by each litter-soil combination, $\delta F^c(t)$ is the average $\delta^{13}C$ of the $CO_2$ respired by the control soil, and $\delta C_l$ is the $\delta^{13}C$ of each litter type. These data were used to calculate cumulative soil-derived C efflux via numerical integration and, for each litter type, average soil-derived C efflux was subtracted from total cumulative $CO_2$-C loss to determine cumulative litter-derived $CO_2$-C loss.

Carbon budget microcosms were harvested on days 30 and 185 to track litter-derived C into mineral-associated SOC at an early and intermediate stage of decomposition. To do this, we used a size fractionation procedure[71,72] modified to minimize the recovery of soluble leaf litter compounds or dissolved organic matter in the mineral-associated SOC fraction. For each sample, we first added 30 mL deionized water, gently shook by hand to suspend all particles, and then centrifuged (2500 rpm) for 10 min. Floating leaf litter was carefully removed, dried for 48 h at 60 °C, and weighed; and the clear supernatant was discarded to remove the dissolved organic matter. The remaining sample was dispersed in 5% (w/v) sodium hexametaphosphate for 20 h on a reciprocal shaker and then washed through a 53 μm sieve. The fraction retained on the sieve was added to the floating leaf litter sample and collectively referred to as particulate SOC, while the fraction that passed through the sieve was considered the mineral-associated SOC. Both fractions were dried, ground, and weighed; and analyzed for C concentrations and $\delta^{13}C$ values on an elemental combustion system (Costech ECS 4010, Costech Analytical Technologies, Valencia, CA, USA) as an inlet to an IRMS. As above, litter-derived C in the particulate and mineral-associated SOC was determined as follows:

$$\frac{C_s^l(t)}{C_s(t)} = \frac{\delta C_s(t) - \delta C_c(t)}{\delta C_l - \delta C_c(t)} \tag{2}$$

where $C_s(t)$ is the total particulate or mineral-associated SOC content in the sample at time ($t$), $C_s^l(t)$ is the litter-derived C in the soil, $\delta C_s(t)$ is the measured $\delta^{13}C$ value for each soil fraction, $\delta C_c(t)$ is the average $\delta^{13}C$ for each fraction in control samples, and $\delta C_l$ is the $\delta^{13}C$ of each litter type. In a few cases, mineral-associated $\delta^{13}C$ was slightly less negative in the treatment than in the control soil. In these cases, litter-derived mineral-associated SOC was considered zero.

Total litter-derived SOC at each harvest date was calculated by subtracting the cumulative litter $CO_2$-C from initial added litter C. The difference between this value and the sum of litter-derived particulate and mineral-associated SOC was considered the residual pool which we assume mostly represents water-extractable dissolved organic matter.

**Microbial biomass dynamics during incubation**. Sample batches were harvested at days 15 and 100 to capture early- and intermediate-term microbial biomass responses to litter treatments. These times were selected to correspond with the middle of early and intermediate C budget microcosm incubations. We quantified microbial biomass as well as MGR, CUE, and MTR using $^{18}O$-labeled water[73,74] as in Geyer et al.[75].

Microbial biomass C (MBC) was determined on two ~2 g subsamples using a standard chloroform fumigation extraction[76]. One subsample was immediately extracted in 0.5 M $K_2SO_4$ and one was fumigated for 72 h before extraction. After shaking for 1 h, extracts were gravity filtered through a Whatman No. 40 filter paper, and filtrates were analyzed for total organic C using the method of Bartlett and Ross[77] as adapted by Giasson et al.[78]. The difference between total organic C in the fumigated and unfumigated subsamples was used to calculate MBC (extraction efficiency $K_{EC} = 0.45$).

To determine MGR, CUE, and MTR, we first pre-incubated two 0.5 g soil subsamples (one treatment and one control) for 2 d at 24 °C. Prior to this pre-incubation, samples were allowed to evaporate down to $53 \pm 6\%$ (mean, sd) water-holding capacity. After the pre-incubation, water was injected with a 25 μL syringe to bring each sample to 65% water-holding capacity. For one subsample, we used unlabeled deionized water. For the second subsample, enriched $^{18}O$-water (98.1 at %; ICON Isotopes) was mixed with unlabeled deionized $H_2O$ to achieve approximately 20 at% of $^{18}O$ in the final soil water. Each sample was placed in a centrifuge tube (modified for gas sampling), flushed with $CO_2$-free air, and incubated for 24 h. Headspace $CO_2$ was then measured, and samples were flash frozen in liquid $N_2$ and stored at −80 °C until DNA extraction.

DNA was extracted from each sample using a DNA extraction kit (Qiagen DNeasy PowerSoil Kit, Venlo, Netherlands) following the protocol described in Geyer et al. (2019) which sought to maximize the recovery of DNA. The DNA concentration was determined fluorometrically using a Quant-iT PicoGreen dsDNA Assay Kit (Invitrogen). DNA extracts (80 μL) were dried at 60 °C in silver capsules spiked with 100 μL of salmon sperm DNA (42.5 ng μL$^{-1}$), to reach the

oxygen detection limit, and sent to the UC Davis Stable Isotope Facility for quantification of $\delta^{18}O$ and total O content.

Microbial growth rate (MGR) was calculated following Geyer et al. (2019). Specifically, atom % of soil DNA O (at% $O_{DNA}$) was determined using the two-pool mixing model:

$$at\% O_{DNA} = \frac{[(at\% O_{DNA+ss} \times O_{DNA+ss}) - (at\% O_{ss} \times O_{ss})]}{O_{DNA}} \tag{3}$$

where at% is the atom % $^{18}O$ and $O_{DNA+ss}$, $O_{DNA}$, and $O_{ss}$ are the concentration of O in the whole sample, soil DNA, and salmon sperm, respectively. Atom percent excess of soil DNA oxygen (APE $O_{soil}$) was calculated as the difference between at% $O_{DNA}$ in the treatment and control samples. Total microbial growth in terms of O (Total O; μg) was estimated as:

$$Total\ O = \frac{O_{soil} \times APE\ O_{soil}}{at\% soil\ water} \tag{4}$$

where at% soil water is the atom % $^{18}O$ in the soil water. MGR in terms of C (μg C g$^{-1}$ soil d$^{-1}$) was calculated by applying conversion mass ratios of oxygen:DNA (0.31) and MBC:DNA for each sample, dividing by the soil mass, and scaling by the incubation time. Assuming uptake rate (Uptake) is equal to the sum of MGR and respiration, CUE and MTR were calculated by the following equations.

$$CUE = \frac{MGR}{Uptake} \tag{5}$$

$$MTR = \frac{MGR}{MBC} \tag{6}$$

**Data analysis for microcosm experiment**. Litter decay constants were calculated for each species using litter-derived $CO_2$-C values to estimate litter mass loss over time. After it was determined that a single exponential decay model provided a poor fit, we fit litter decomposition data using the double exponential decay model:

$$y = se^{-k_1 t} + (1-s)e^{-k_2 t} \tag{7}$$

where $s$ represents the labile or early stage decomposition fraction that decomposes at rate $k_1$, and $k_2$ is the decay constant for the remaining late stage decomposition fraction.

To reduce the dimensionality of litter quality and microbial indicators, indices were derived by principal component analysis (PCA; Fig. S1A, B) using the 'prcomp' function in R. The first axis of a PCA of decomposition parameters ($s$, $k_1$, and $k_2$) and litter chemical properties (soluble and AUR contents; AUR-to-N and C-to-N ratios; and the lignocellulose index [LCI]) was taken as a litter quality index. Whereas this index highly correlated with indicators of C quality (AUR, soluble content, and LCI), the second axis of this PCA correlated with C:N and AUR:N and was therefore taken as a second litter quality index representing variation in N concentration. The first axis of a PCA of MGR, CUE, and MTR was taken as a microbial physiological trait index.

Bivariate relationships were examined using simple linear regressions on average species values at each harvest ($n = 16$). To examine relationships between microbial physiological traits and mineral-associated SOC, data from the early-term (day 15) and intermediate-term (day 100) microbial harvest were matched with early-term (day 30) and intermediate-term (day 185) C budget microcosms, respectively. In addition to examining total mineral-associated SOC formation, we also estimated the efficiency of litter C transfer into the mineral-associated SOC pool as the fraction of lost litter C (i.e., litter C lost as $CO_2$, recovered in the mineral-associated SOC fraction, or in the residual pool) retained in the mineral-associated SOC. Path analyses were used to evaluate the hypothesis that microbial physiological traits mediate the effect of litter quality on mineral-associated SOC formation and mineral-associated and particulate SOC decay. We hypothesized that the litter quality index would be positively associated with the microbial physiological trait index (representing faster and more efficient microbial growth) and microbial physiological traits would, in turn, be positively associated with the rate and efficiency of mineral-associated SOC formation. We expected that this mediating pathway would reduce the direct relationship between litter quality and SOC. This analysis was conducted using the LAVAAN package[79] to run path analyses for total litter-derived mineral-associated SOC, mineral-associated SOC formation efficiency, and soil-derived mineral-associated and particulate SOC for both early and intermediate stage harvests. All analyses were performed using R version 3.5.2.

**Field study design and soil sampling**. We worked in the Smithsonian's Forest Global Earth Observatory (ForestGEO) network[80] in six mature U.S. temperate forests varying in climate, soil properties, and tree community composition (Fig. 1a): Harvard forest (HF; 42°32′N, 72°11′W) in North-Central Massachusetts, Lilly-Dickey Woods (LDW; 39°14′N, 86°13′W) in South-Central Indiana, the Smithsonian Conservation Biology Institute (SCBI; 38°54′N, 78°9′W) in Northern Virginia, the Smithsonian Environmental Research Center (SERC; 38°53′N, 76°34′W) on the Chesapeake Bay in Maryland, Tyson Research Center (TRC; 38°31′N, 90°33′W) in Eastern Missouri, and Wabikon Lake Forest (WLF; 45°33′N, 88°48′W) in Northern Wisconsin, USA. Land use history across the six sites consisted mostly

of timber harvesting which ceased in the early 1900s. Soils are mostly Oxyaquic Dystrudepts at HF, Typic Dystrudepts and Typic Hapludults at LDW, Typic Hapludalfs at SCBI, Typic or Aquic Hapludults at SERC, Typic Hapludalfs and Typic Paleudalfs at TRC, and Typic and Alfic Haplorthods at WLF. Further site details are reported in Table S5.

Each site contains a rich assemblage of co-occurring arbuscular mycorrhizal (AM)- and ectomycorrhizal (ECM)-associated trees (Table S6), which we leveraged to generate environmental gradients in factors hypothesized to predict microbial physiological traits within each site. Specifically, the dominance of AM vs. ECM trees within a temperate forest plot has been shown to be a strong predictor of soil pH, C:N, inorganic N availability, and litter quality[52–54]. We established nine $20 \times 20$ m plots in each of our six sites in Fall 2016 ($n = 54$) distributed along a gradient of AM- to ECM-associated tree dominance. Plots were selected to avoid obvious confounding environmental factors. Where possible, we established our nine-plot gradient in three blocks (<1 ha), each containing an AM, ECM, and mixed dominance plot. Plots were generally located on upland areas except for TRC where all plots were located on toeslopes due to a lack of AM trees in upland areas. At HF, we established six of the nine plots outside the boundaries of the ForestGEO plot due to a lack of appropriate AM-ECM mixtures. At LDW, we used a random subset of previously established $15 \times 15$ m plots[81]. Basal area of living trees was determined using the most recent ForestGEO census (within the last 5 years) or by conducting our own inventory. ECM-dominance was quantified as the percentage of ECM-associated basal area relative to the total plot basal area.

Four soil cores (5 cm-diameter) were collected in July 2017 from each plot to a depth of 5 cm. A thin O horizon (Oe and Oa) was sometimes present and was collected as part of the 5 cm soil core. However, there was often a thick O horizon (>5 cm) at HF, which was removed before coring. Samples were also collected at 5–15 cm depth for soil texture analysis. We sampled from an inner $10 \times 10$ m square in each plot to avoid edge effects. All samples from the same plot were composited, sieved (2 mm), picked free of roots, subsampled for gravimetric moisture (105 °C), and air-dried, or refrigerated (4 °C) until analysis for microbial physiological variables and N availability.

**Soil properties.** We determined several physicochemical properties known to predict mineral-associated SOC. We measured soil pH (8:1 ml 0.01 M CaCl₂:g soil) and soil texture using a benchtop pH meter and a standard hydrometer procedure[82], respectively. Organic matter content was high in some upper surface soils, so plot-level soil texture was determined from 5 to 15 cm depth samples. We quantified oxalate-extractable Al and Fe pools ($Al_{ox}$ and $Fe_{ox}$) in all soil samples as an index of poorly crystalline Al- and Fe-oxides[83], which is one of the strongest predictors of SOM content in temperate forests[84]. Briefly, we extracted 0.40 g dried, ground soil in 40 mL 0.2 M NH₄-oxalate at pH 3.0 in the dark for 4 h before gravity filtering and refrigerating until analysis (within 2 w) on an atomic-adsorption spectrometer (Aanalyst 800, Perkin Elmer, Waltham, MA, USA), using an acetylene flame and a graphite furnace for the atomization of Fe and Al, respectively.

We quantified potential net N mineralization rates as an index of soil N availability. One 5 g subsample per plot was extracted immediately after processing by adding 10 mL 2 M KCl, shaking for 1 h, and filtering through a Whatman No. 1 filter paper after centrifugation at 3000 *rpm*. A second subsample from each plot was incubated under aerobic conditions at field moisture and 23 °C for 14 d before extraction. Extracts were frozen (−20 °C) until analysis for NH₄⁺-N using the salicylate method and for NO₃⁻-N plus NO₂⁻-N after a cadmium column reduction on a Lachat QuikChem 8000 flow Injection Analyzer (Lachat Instruments, Loveland, CO, USA). Potential net N mineralization rates (mg N g dry soil⁻¹ d⁻¹) were calculated as the difference between pre- and post-incubation inorganic N concentrations.

**Microbial biomass dynamics in field plots.** Microbial biomass carbon and microbial physiological traits were quantified within 10 days of collection as described above, with four minor differences. First, 30 g soil subsamples were covered with parafilm and pre-incubated for 2 d near the field soil temperature measured at the time of sampling (16.5 °C for WLF and HF, and 21.5 °C for LDW, TRC, SCBI, and SERC). Second, for CO₂ analysis, samples were placed in a 61 mL serum vial crimped with a rubber septum. Third, DNA concentrations were determined using a Qubit dsDNA BR Assay Kit (Life Technologies) and a Qubit 3.0 fluorometer (Life Technologies). Fourth, 14.5 g subsamples were used for microbial biomass analysis.

**Soil organic matter characterization in field plots.** Mineral-associated SOC was quantified as in the microcosm experiment, but without a pre-fractionation leachate removal step. We additionally measured soil amino sugar concentrations to estimate microbial necromass contributions to SOM. Amino sugars are useful microbial biomarkers because they are found in abundance in microbial cell walls, but are not produced by higher plants and soil animals[19]. Moreover, amino sugars can provide information on the microbial source of necromass. For example, glucosamine (Glu) is produced mostly by fungi whereas muramic acid (MurA) is produced almost exclusively by bacteria[61,85]. Amino sugars were extracted, purified, converted to aldononitrile acetates, and quantified with myo-inositol as in Liang et al.[86]. We used the concentrations of Glu and MurA to estimate total,

fungal, and bacterial necromass soil C using the empirical relationships reported in Liang et al.[8].

$$Bacterial\ necromass\ C = MurA \times 45 \quad (8)$$

$$Fungal\ necromass\ C = (mmol\ GluN - 2 \times mmol\ MurA) \times 179.17 \times 9 \quad (9)$$

**Leaf litter and fine roots in field plots.** In Fall 2017, we collected leaf litter on two sample dates from four baskets deployed in the inner $10 \times 10$ m of each plot. Litter was composited by plot, dried (60 °C), sorted by species, weighed, and ground. We performed leaf litter analyses on at least three samples of each species at each site —unless a species was only present in one or two plots— to get a site-specific mean for each species. Some non-dominant species were not included in these analyses because an insufficient amount of material was collected. Fine roots (<2 mm) were collected at the time of soil sampling (at LDW, TRC, and WLF) or the following summer (HF, SERC, SCBI) due to issues with sample processing.

Leaf litter and bulk root samples were analyzed for C and N content. For leaf litter, we conducted a sequential extraction as above (sensu Moorhead & Reynolds[87]) to determine the soluble fraction using hot water and ethanol, the AUR—the insoluble ash-free fraction that resisted degradation by a strong acid— which contains lignin and is commonly found to inhibit leaf litter decay[40]. The LCI was estimated by dividing the AUR fraction by the total non-soluble ash-free fraction. Leaf litter trait values were calculated for each plot as the average of site-specific mean values for each species in the plot weighted by proportion of total plot basal area.

**Data analysis for field study.** As above, a litter quality index and microbial physiological trait index was calculated via PCA for each plot (Fig. S1C, D). We note, however, that decomposition parameters were not available for the litter quality PCA in the field study. We evaluated the extent to which microbial physiological variables were predicted by litter quality, N availability, stoichiometry, C availability, pH, or root biomass. We first examined variability in indicators of these biogeochemical drivers (i.e., the litter quality index, net N mineralization, soil C:N, dissolved organic carbon (DOC), soil pH, and fine root biomass) and evaluated whether they correlated with our mycorrhizal gradients. Linear mixed models were fit to the microbial physiological trait index using site as a random factor and the above indicators as fixed factors.

We also used linear mixed models to evaluate whether microbial physiological variables predicted significant variation in mineral-associated or necromass-derived C in the context of other hypothesized or known controls on mineral-associated SOC. In addition to the litter quality and microbial physiological trait indices, we included clay content, pH, $Fe_{ox}$, $Al_{ox}$, and fine root biomass as fixed factors. We fit models to mineral-associated SOC—on a total basis (g soil⁻¹) and as a proportion of soil C (mineral-associated C/total soil C)—as well as total, fungal, and bacterial necromass. Models were selected to avoid highly correlated predictors (i.e., $r > 0.5$). $Fe_{ox}$ and $Al_{ox}$ were correlated above this threshold and final models were selected to contain only $Fe_{ox}$ based on AIC. Residuals were screened for normality (Shapiro-Wilk), heteroscedasticity (visual assessment of residual plots), and influential observations (Cook's D). Based on this, MGR, MTR, and mineral-associated SOC were natural log transformed. For all mixed models, we centered and standardized predictors (i.e., z-transformation) so that the slopes and significance levels of different predictors could be compared to one another on the same axis[88].

## Data availability

The data generated in this study have been deposited in the ESS-DIVE archive[89] (https://data.ess-dive.lbl.gov/view/; https://doi.org/10.15485/1835182).

## Code availability

Analysis code is available as part of the ESS-DIVE data package[89] (https://data.ess-dive.lbl.gov/view/; https://doi.org/10.15485/1835182).

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

## Acknowledgements
This work was supported by a National Science Foundation Doctoral Dissertation Improvement Grant (DEB-1701652; M.E.C. and R.P.P.), the U.S. Department of Energy Office of Biological and Environmental Research (DOE-BER), Terrestrial Ecosystem Science Program (DESC0016188; E.R.B. and R.P.P.), an Indiana University Research and Teaching Preserve (IURTP) Student Grant (M.E.C.), the Smithsonian Center for Tropical Forest Science—ForestGEO, the Oak Ridge National Laboratory (ORNL) Terrestrial Ecosystem Science, Science Focus Area, funded by DOE-BER (M.E.C.), and the National Natural Science Foundation of China (31930070; C.L.). We thank Elizabeth Huenupi and members of the Phillips, Brzostek, and Frey labs (Corben Andrews, Kelly Fox, Peyton Joachim, Naomi Reibold, Madison Barney, Rachel Zeunik, Mark Sheehan, Kara Allen, Joe Carrara, Nanette Raczka, and others) for assistance in the field and lab. We also thank Peter Sauer, Erica Elswick, Ryan Mushinski, and Brent Lemkuhl for assistance with sample analyses; Ron Turco for facilitating soil collection; individuals affiliated with the ForestGEO network for facilitating site access (Bill McShea, Dave Orwig, Sean McMahon, Michael Chitwood, Jonathan Myers, and Amy Wolf); and Adrienne Keller and Steve Kannenberg for feedback on earlier presentations of this work. LDW is part of the IURTP. ORNL is managed by UT-Battelle, LLC, for the U.S. DOE under contract DE-AC05-100800OR22725.

## Author contributions
M.E.C. and R.P.P. conceived the project with contributions from A.S.G. and E.R.B. on the lab and field studies, respectively. M.E.C. and K.V.B. carried out the lab experiment with input from K.M.G. M.E.C. and E.R.B. contributed to field sampling. M.E.C., K.M.G., and C.L. analyzed field samples. M.E.C. analyzed the data and wrote the first draft of the paper. All authors contributed to the conceptual development of the paper and provided feedback on the final draft.

## Competing interests
The authors declare no competing interests.
