## [Peer Review File · Nature Communications]

REVIEWER COMMENTS

Reviewer #1 (Remarks to the Author):

Summary and Contribution:

This study focuses on the testing the hypothesis that a large proportion of soil carbon is derived from microbial necromass and that the contribution of microbial necromass to soil carbon is greater when the litter is of high quality. Contrary to the hypothesis, the results indicated a negative relationship between metrics for microbial growth, carbon use efficiency and turnover, and mineral stabilized soil organic carbon in temperate forest ecosystems. The study design is sound, the results were interpreted well and the conclusions are largely robust. I recommend publishing the study with minor revisions.

Major Points:

Title: High quality litter - to a non specialist audience, high 'quality' litter is a vague and poorly defined term. Update to something more clearly descriptive for this journal with a broad readership. Highly decomposable litter? I'm not sure what the best term to substitute would be, but 'high quality litter' seems too niche in this case.

The title also indicates that this result may be universally true, but the study was conducted with plant litters from temperate forests. The study should be couched in results for temperate forests and at least a paragraph of the discussion should be dedicated to the differences in characteristics of e.g. grasslands or agriculture systems and how likely the same mechanisms are to be in play, or if the necromass stabilization hypothesis would be a greater or smaller component contributing to soil carbon stocks in those ecosystems.

Line 39: offer specifics of the 'broadened view' it's unsatisfying to say what it is not - can you offer what the data indicate the primary drivers of SOC persistence in this ecosystem may be?

Line 114 : One of the most common ways of quickly assessing the decomposability of a substrate is the C:N ratio, but those values were not included in Table S1 (but are analyzed in the PCA for Figure S1). The AUR:N is included which should be an approximation of this, but without the C:N values it's hard to understand what the AUR:N captures that the C:N does not, and also hard to reference the range of litter characteristics when compared with something known to be highly decomposable, versus something that isn't. Please add the C:N values for the litters in Table S1 to make it easier to put the data in a broader context.

Minor Points

Line 45: is it really microbial products? Do we really know this? Ref 5 is a conceptual paper - are there better refs for this? More compelling evidence?

Line 115: don't follow how tighter relationship between LQI and MPT later in the incubation means faster depletion of labeled substrates in low-quality litters - is it really faster, or is there just less? Is the rate dependent upon the amount or on the microbes?

Line 164: what are 'these effects'

Line 169: Couldn't enzymes and EPS work in opposition to one another? I.e. EPS would stabilize C and enzymes would reduce the amount of stabilized C.

Reviewer #2 (Remarks to the Author):

The authors of "High quality litters enhance soil carbon, not through microbial physiological traits" present

results from two studies focused on the role of microbial traits (CUE, MGR, and turnover) on mineral associated soil organic carbon. In addition to these microbial traits are several other mechanisms that may play an important role in MA-SOC accumulation (presented in text and Figure 1). The results from both microcosm and field studies indicate that the necromass stabilization hypothesis may not contribute greatly to SOC accumulation. The results are well described and discussed. Quantifying the role of different mechanisms in the soil can be difficult and the combination of methods used in these studies have helped clarify that microbial physiological traits may not be the dominant control on mineral stabilization of plant inputs

REVIEWER COMMENTS

Reviewer #1 (Remarks to the Author):

Summary and Contribution:

This study focuses on the testing the hypothesis that a large proportion of soil carbon is derived from microbial necromass and that the contribution of microbial necromass to soil carbon is greater when the litter is of high quality. Contrary to the hypothesis, the results indicated a negative relationship between metrics for microbial growth, carbon use efficiency and turnover, and mineral stabilized soil organic carbon in temperate forest ecosystems. The study design is sound, the results were interpreted well and the conclusions are largely robust. I recommend publishing the study with minor revisions.

Thank you for your support of our paper and for your thoughtful and helpful criticisms. We have made changes in response to each of your comments, resulting in a manuscript that is more accessible to generalist readers and more specific to temperate forests. Please see our detailed responses below.

Major Points:

Title: High quality litter - to a non specialist audience, high 'quality' litter is a vague and poorly defined term. Update to something more clearly descriptive for this journal with a broad readership. Highly decomposable litter? I'm not sure what the best term to substitute would be, but 'high quality litter' seems too niche in this case.

1) We agree and have updated the title with the more intuitive term “fast-decaying plant litter”. We also now define the term “high-quality plant litter” at first mention in the abstract (line 31).

New title: Fast-decaying plant litter enhances soil carbon in temperate forests, but not through microbial physiological traits

Lines 29-32: Recent hypotheses suggest that microbial necromass production links plant inputs to SOC accumulation, with high-quality (i.e. rapidly decomposing) plant litter promoting microbial carbon use efficiency, growth, and turnover leading to more mineral stabilization of necromass.

The title also indicates that this result may be universally true, but the study was conducted with plant litters from temperate forests. The study should be couched in results for temperate forests and at least a paragraph of the discussion should be dedicated to the differences in characteristics of e.g. grasslands or agriculture systems and how likely the same mechanisms are to be in play, or if the necromass stabilization hypothesis would be a greater or smaller component contributing to soil carbon stocks in those ecosystems.

2) The reviewer makes an excellent point. We have added a paragraph to make this point clear and discuss how the importance of necromass stabilization could vary across systems (lines 305-314). Additionally, we have added the phrase “in temperate forests” to the title, abstract (lines 38-39), and main text (lines 89-91 and 161-162) to avoid implying that our results can be completely extrapolated to all systems.

Lines 305-314: While we found that microbial growth, efficiency, and turnover are not adequate predictors of mineral-associated SOC formation in surface soils of temperate forests, necromass stabilization on mineral surfaces could still be an important mechanism of SOC storage. This mechanism may explain more variation in other systems (e.g. croplands or grasslands) or in deeper soils, where necromass accounts for a larger proportion of total SOC⁹. For example, agricultural soils tend to be more bacterially dominated⁹, potentially leading to stronger interactions between microbial necromass and soil minerals (as described above). This may lead to a stronger association between microbial physiological traits and mineral-associated SOC in cropped systems³⁰. Future sampling campaigns are needed to understand how SOC formation pathways vary across different systems.

Line 39: offer specifics of the ‘broadened view’ it’s unsatisfying to say what it is not - can you offer what the data indicate the primary drivers of SOC persistence in this ecosystem may be?

3) We agree with the reviewer. Our data point to several important factors which we now list in the abstract.

Lines 37-41: We suggest that microbial necromass production is not the primary driver of SOC persistence in temperate forests. Factors such as microbial necromass origin, alternative SOC formation pathways, priming effects, and soil abiotic properties can strongly decouple microbial growth, efficiency, and turnover from mineral-associated SOC.

Line 114 : One of the most common ways of quickly assessing the decomposability of a substrate is the C:N ratio, but those values were not included in Table S1 (but are analyzed in the PCA for Figure S1). The AUR:N is included which should be an approximation of this, but without the C:N values it's hard to understand what the AUR:N captures that the C:N does not, and also hard to reference the range of litter characteristics when compared with something known to be highly decomposable, versus something that isn't. Please add the C:N values for the litters in Table S1 to make it easier to put the data in a broader context.

4) These values have now been added to Table S1, as the reviewer suggests. In the process of double-checking tables and figures, we also discovered that some chemistry values had been reversed for *Fagus grandifolia* and *Fraxinus americana*. This error has been corrected and did not affect any analyses.

Species	C:N	% N	% Soluble	% AUR	LCI	AUR:N	s	k ₁	k ₂	% C loss
Acer rubrum	89.7 (4.6)	0.52 (0.03)	61.7 (0.2)	7.0 (2.4)	0.19 (0.06)	13.1 (4.1)	0.31 (0.01)	0.15 (0.01)	0.0036 (0.0002)	62.2 (0.7)
Acer saccharum	49.6 (2.6)	0.89 (0.05)	50.4 (0.7)	12.4 (0.2)	0.27 (0.00)	14.1 (0.7)	0.27 (0.02)	0.14 (0.03)	0.0038 (0.0004)	60.8 (3.8)
Asimina triloba	21.7 (0.4)	2.09 (0.04)	39.7 (0.2)	15.6 (0.4)	0.26 (0.01)	7.4 (0.3)	0.27 (0.01)	0.11 (0.01)	0.0006 (0.0001)	34.1 (0.9)
Carya cordiformis	42.7 (1.8)	1.05 (0.05)	32.1 (1.4)	20.7 (0.8)	0.31 (0.01)	19.9 (0.9)	0.27 (0.01)	0.09 (0.01)	0.0012 (0.0001)	39.9 (0.3)
Carya glabra	50.2 (0.8)	0.89 (0.02)	29.1 (0.6)	22.7 (0.6)	0.32 (0.01)	25.5 (1.2)	0.27 (0.01)	0.07 (0.00)	0.0010 (0.0001)	38.4 (0.7)
Carya ovata	47.2 (3.5)	0.95 (0.08)	34.1 (1.4)	20.6 (0.4)	0.32 (0.00)	22.1 (2.0)	0.32 (0.03)	0.07 (0.01)	0.0015 (0.0004)	48.0 (4.1)
Fagus grandifolia	49.3 (1.0)	0.92 (0.02)	26.6 (0.4)	22.9 (0.2)	0.25 (0.01)	24.9 (0.6)	0.22 (0.01)	0.05 (0.01)	0.0017 (0.0001)	42.5 (1.2)
Fraxinus americana	39.8 (1.1)	1.04 (0.08)	41.7 (0.8)	14.5 (0.9)	0.32 (0.00)	14.1 (1.1)	0.29 (0.01)	0.15 (0.01)	0.0014 (0.0002)	43.4 (1.7)
Liriodendron tulipifera	57.6 (2.1)	0.78 (0.03)	52.0 (0.7)	13.7 (0.4)	0.29 (0.00)	17.7 (1.1)	0.23 (0.01)	0.16 (0.01)	0.0007 (0.0001)	31.6 (0.6)
Nyssa sylvatica	64.2 (1.2)	0.71 (0.02)	54.4 (1.1)	9.9 (0.2)	0.22 (0.01)	14.1 (0.3)	0.23 (0.01)	0.13 (0.01)	0.0033 (0.0001)	55.2 (0.7)
Quercus alba	57.7 (0.7)	0.82 (0.01)	50.3 (1.4)	15.0 (0.4)	0.31 (0.00)	18.2 (0.4)	0.36 (0.08)	0.03 (0.01)	0.0011 (0.0008)	47.4 (3.2)
Quercus prinus	74.9 (1.2)	0.63 (0.01)	48.0 (1.0)	17.3 (0.2)	0.35 (0.01)	27.5 (0.7)	0.22 (0.01)	0.06 (0.01)	0.0017 (0.0001)	42.5 (1.1)
Quercus rubra	71.0 (4.8)	0.69 (0.05)	35.9 (1.2)	24.5 (0.3)	0.39 (0.01)	36.1 (2.7)	0.23 (0.01)	0.07 (0.01)	0.0015 (0.0001)	40.3 (1.8)
Quercus velutina	61.6 (4.8)	0.79 (0.06)	40.4 (0.7)	21.8 (0.4)	0.37 (0.00)	28.1 (2.0)	0.22 (0.01)	0.06 (0.01)	0.0012 (0.0001)	36.9 (1.4)
Sassafras albidum	57.7 (3.6)	0.83 (0.04)	54.0 (0.9)	15.4 (0.7)	0.34 (0.02)	18.6 (1.9)	0.16 (0.01)	0.16 (0.02)	0.0011 (0.0001)	30.2 (0.7)
Tilia americana	27.5 (0.8)	1.69 (0.06)	42.7 (1.2)	23.9 (2.2)	0.42 (0.03)	14.2 (1.6)	0.22 (0.01)	0.11 (0.01)	0.0010 (0.0001)	33.9 (1.5)

Minor Points

Line 45: is it really microbial products? Do we really know this? Ref 5 is a conceptual paper - are there better refs for this? More compelling evidence?

5) Thank you. We have added two more recent studies (see below) that provide direct evidence for the abundance of microbial products in association with soil minerals. There were a number of studies and reviews in the early 2000s, but we feel that ref 5 nicely synthesizes this work and have chosen to keep this citation.

Bradford, M. A., Keiser, A. D., Davies, C. A., Mersmann, C. A., & Strickland, M. S. (2013). Empirical evidence that soil carbon formation from plant inputs is positively related to microbial growth. *Biogeochemistry*, *113*(1–3), 271–281. <https://doi.org/10.1007/s10533-012-9822-0>

Miltner, A., Bombach, P., Schmidt-Brücken, B., & Kästner, M. (2012). SOM genesis: Microbial biomass as a significant source. *Biogeochemistry*, *111*(1–3), 41–55. <https://doi.org/10.1007/s10533-011-9658-z>

Line 115: don't follow how tighter relationship between LQI and MPT later in the incubation means faster depletion of labeled substrates in low-quality litters - is it really faster, or is there just less? Is the rate dependent upon the amount or on the microbes?

6) This is a good catch. As the reviewer indicates, we meant to hypothesize here that the stronger effect of litter quality on microbial physiological traits later in the incubation was a function of the smaller—and therefore more rapidly depleted—pool of labile substrates in low-quality litters. We have clarified the text.

Line 117-119: This effect was especially pronounced during intermediate stages of decomposition suggesting a depletion of **the smaller pool of** labile substrates in low-quality litters.

Line 164: what are 'these effects'

7) Thank you. We have clarified this sentence.

Lines 167-168: These effects **of litter quality on dissolved SOC** are known to enhance mineral-associated SOC formation⁴².

Line 169: Couldn't enzymes and EPS work in opposition to one another? I.e. EPS would stabilize C and enzymes would reduce the amount of stabilized C.

8) Yes. This is a good point. Enzymes can certainly interact with mineral surfaces and therefore contribute to mineral-associated soil C formation, but they likely have an opposing net effect on soil C relative to other extracellular compounds. For simplicity, we have revised the text to de-emphasize enzymes (which we previously listed as a potentially stabilized extracellular product).

Lines 168-173: Additionally, microbial extracellular products could be important (Fig. 1A), but are overlooked by an emphasis on microbial physiological traits^{33,43}. **Extracellular polymeric substances, stress compounds, and similar products** account for a small proportion of microbial production⁴⁴, but are likely to disproportionately affect mineral-associated SOC given their susceptibility to stabilization on mineral surfaces³⁴.

Reviewer #2 (Remarks to the Author):

The authors of "High quality litters enhance soil carbon, not through microbial physiological traits" present results from two studies focused on the role of microbial traits (CUE, MGR, and turnover) on mineral associated soil organic carbon. In addition to these microbial traits are several other mechanisms that may play an important role in MA-SOC accumulation (presented in text and Figure 1). The results from both microcosm and field

studies indicate that the necromass stabilization hypothesis may not contribute greatly to SOC accumulation. The results are well described and discussed. Quantifying the role of different mechanisms in the soil can be difficult and the combination of methods used in these studies have helped clarify that microbial physiological traits may not be the dominant control on mineral stabilization of plant inputs

Thank you for your support of our manuscript. We were excited to tackle this question with multiple approaches and are glad to receive your positive response.

REVIEWERS' COMMENTS

Reviewer #1 (Remarks to the Author):

The authors have done a great job of addressing all of the suggested minor revisions, I recommend publication in the current form. Nicely done!

REVIEWERS' COMMENTS

Reviewer #1 (Remarks to the Author):

The authors have done a great job of addressing all of the suggested minor revisions, I recommend publication in the current form. Nicely done!

We thank this reviewer for their helpful comments. We are happy to receive such a positive response to our paper and its revisions.